# Brain fog in chronic pain: Protocol for a discourse analysis of social media postings

**Ronessa Dass** ⬥ *, Tara Packham

Rehabilitation Sciences, McMaster University, Hamilton, Ontario, Canada

* dassr5@mcmaster.ca

## Abstract

Brain fog is a phenomenon that is frequently reported by persons with chronic pain. Difficulties with cognition including memory impairments, attentional issues, and cloudiness are commonly described. The current medical literature demonstrates a similar cloudiness: there is no clear taxonomy or nomenclature, no well-validated evaluations and a dearth of effective interventions. To focus our understanding of this complex phenomenon, we will perform a discourse analysis to explore how brain fog is described in public posts on social media. Discursive methodology will generate insights regarding the societal understanding and meanings attributed to brain fog, by sampling perspectives of persons with lived experience, currently underrepresented in the medical literature. It is anticipated that the results of the proposed study will 1) help healthcare professionals better understand the experience of chronic pain-related brain fog and 2) generate hypotheses for future research. To conclude, by incorporating innovative and contemporary methods, this proposed discourse analysis of social media sources will generate nuanced insights, bridging the gap between researchers, health care providers, and persons with lived experience.

**Data Availability Statement:** All relevant data are within the paper and its Supporting Information files.

**Funding:** The author(s) received no specific funding for this work.

## Introduction

Brain fog is a phenomenon that is frequently reported by persons with chronic pain [1,2]. It is commonly described as difficulties with cognition including memory impairments, attentional issues, cloudiness, and more [3–6]. The term 'brain fog' was initially developed by persons with lived experiences (PWLE), primarily through online discussions [7]. In the medical literature, it remains poorly understood and does not have an agreed definition [7,8]. To help resolve this issue, we performed a scoping review of the medical literature to inform a working definition of brain fog [8]. The definition positions brain fog as subjective state of cognitive dysfunction that varies across and within individuals, impacting participation in daily activities [8]. To expand our understanding of this complex phenomenon, we will perform a discourse analysis by searching text-based postings on two social media sources, Twitter and Facebook, to explore how brain fog is described in public discourses on social media.

Discourse analyses are beneficial in understanding how the discussion around a particular topic (e.g., vocabulary used, personal anecdotes and opinions) form its societal understanding and meaning [9–12]. Discourses about a particular phenomenon are communication-based

**Competing interests:** The authors have declared that no competing interests exist.

actions that are created by one's experiences and that are continuously evolving to shape an overall understanding of the topic [9–11]. As it establishes an understanding of a topic, a discourse may in turn impact how individuals perceive a topic and engage with it [11,13]. For example, the discourse individuals see about the phenomenon of brain fog may affect how they identify with their condition, how they manage it, and the role it plays in their daily lives [11,13].

Social media platforms are a powerful tool used by stakeholders including PWLE, researchers, and health care professionals to promote awareness and understanding of different conditions [14,15]. Many persons with chronic pain use social media platforms for health advocacy by sharing their experiences and receiving support from others [14]. Persons with chronic pain may share strategies to help others understand their experiences, communicate with health care providers, and identify management strategies [14].

The use of social media data also has many benefits for research [15]. For example, social media data is not limited by geographic boundaries full in monitoring emerging public health trends to aid in estimating attributes, causes, and potential interventions [15,16].

Though social media data may be inherently biased, and at times inaccurate, the discourse often has real life implications [14,15]. Persons with chronic pain may adopt the information they see online and may be influenced to alter their beliefs about their conditions [14,15]. For example, individuals may use a particular intervention they see online or may describe their condition using words from social media posts [14,15]. However, exploration of the vast amount of information and available on social media may have negative consequences, such as cognitive overload [17,18]. This can be problematic in the context of brain fog in which cognitive load may be limited [7]. Increased cognitive load may also decrease one's ability to critically evaluate and verify the information they are receiving [17]. A study investigating social media fatigue during the early stages of the COVID pandemic stated that social media fatigue decreased participants fact-checking behaviour and led to information avoidance: and this avoidance had potential negative health consequences [18]. The study concluded that examining how target populations use social media for health concerns can help researchers develop effective dissemination strategies for those groups [18]. Given that brain fog is a phenomenon that is predominantly discussed in online discourse on social media platforms [7], a deductive analysis of social media sources is highly relevant. These results may aid in 1) understanding the causes, attributes, and potential interventions for brain fog and 2) bridging the understanding between scientific sources and natural discussions.

Therefore, the overarching objective of this study will be to explore how brain fog in chronic pain is described in public discourses on several text-based social media platforms. Our specific objectives include:

1. To explore if chronic pain related brain fog is described differently pre and post the COVID-19 pandemic.

2. To explore if brain fog described differently on social media across the social groups of researchers, health care professionals, and persons with lived experience.

3. To synthesize how the symptoms of brain fog are described in public discourses on social media.

4. To identify management strategies used by persons with brain fog or suggested by others.

5. To contrast our working definition of brain fog with descriptions posted on social media.

## Methods

### Search strategy

An initial formative search of social media databases was iteratively performed to select relevant key terms and the appropriate databases.

For the purpose of this study, text-based social media posts on Twitter and Facebook will be used as data sources. Twitter and Facebook have been selected as they: 1) are text centric data bases with open access, 2) are widely used by our social groups of interest, 3) have the opportunity for discourse amongst users, 4) have a wide range of data source type (e.g., text, videos, and infographics), and 5) have robust search engines [16,19,20]. Search terms "#brainfog chronic pain," "brain fog chronic pain," "#brainfog chronic pain" and "brain fog #chronicpain" will be used to ensure that the search is comprehensive and does not miss any potential sources which merit inclusion [21]. To maintain the feasibility of the study and to achieve one of our secondary objectives addressing potential shifts in discourse after the onset of the COVID pandemic, we will be restricting our search from the years 2020–2022 and 2016–2018. The search will be run by the primary researcher (RD) and will run from new purpose-created account with no memberships in any groups or communities, to reduce any ethical concerns [22]. No other filters will be applied. In accordance the Internet Specific Ethical Questions Framework to protect data anonymity [22], searches will occur manually through a public search engine, therefore no private posts, pages, or groups will be searched. All searches will be conducted within the timespan of one week, to ensure feasibility as new posts generate daily on social media platform.

### Types of sources

Sources that will be included

- Public posts in English

- Posts discussing brain fog in adults with a painful chronic condition (e.g., chronic pain, chronic musculoskeletal pain)

- Posts must be explicitly referring to chronic pain, or refers to chronic pain in the profile, or that discusses chronic pain in adjacent thread posts.

- Text posts about or in response to a video or image

  Sources that will be excluded

- Post in a language other than English

- Brain fog related to conditions other than chronic pain (e.g., fibro fog, long COVID, chemo fog)

- Posts focused on children, as they have systematic differences in cognition [23]

- Sources from closed community support groups as these are private spaces or a response to a post from a closed community [22]

- Video or image-based posts (e.g. pictures with ALT text, memes)

- Retweets or duplicated tweets

## Data extraction

All posts will be saved into a data extraction form, which will be located on a password protected device and will only be shared with researchers directly involved with the project [22]., data extraction form was informed by a) Foley's concept analysis framework to support systematic elaboration of the concept of brain fog [24], b) Internet Specific Ethical Questions Framework to protect data anonymity [22], c) the International Classification of Functioning to categorize the impacts described [25], and d) Dass et al's (2023) model of brain fog [8]. The data extraction form will be piloted with the first ten posts and will be modified if needed, through a discussion with both reviewers (see Fig 1 for key constructs to be included in extraction; RD and TP; [21]). Exact quotes will be stored on the data extraction form, however, we will use synthetic or paraphrased quotes in the official manuscript to protect anonymity [14,26,27]. Additionally, we will also collect demographic information on the stakeholder type [PWLE, researcher, or health care professional; 14,26,27]. We will manually label and classify stakeholders using information from the user profile. Data extraction will be completed manually by two researchers, as a trustworthiness strategy. Since we are using sources from social media, underlying findings will not be made available to maintain the anonymity of sources [22]. All extracted information will be deleted post manuscript completion.

## Data analysis and presentation

The data derived from this review will be used to deductively underpin both a mapping review and concept analysis. In the mapping analysis, we will compare and contrast 1) how chronic pain related brain fog is described pre and post the COVID-19 pandemic and 2) how chronic pain related brain fog is described across 3 different stakeholder groups: researchers, health

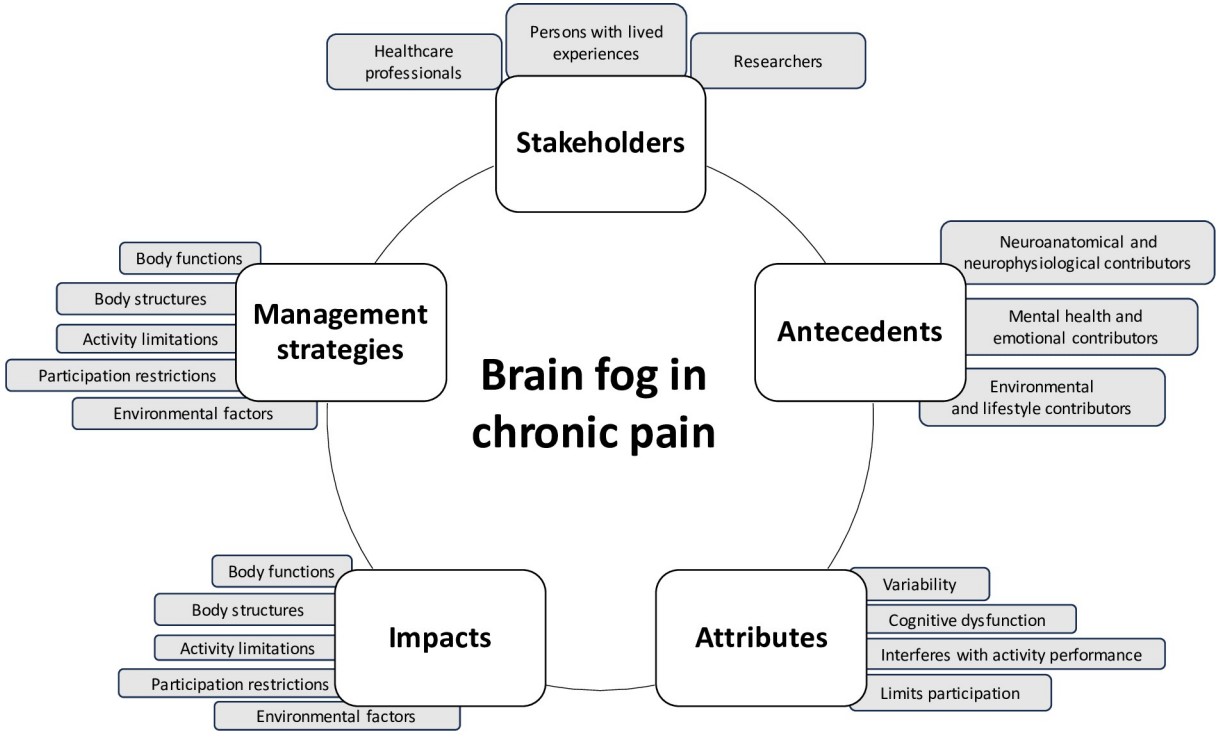

**Fig 1. Approximately here: Key constructs for data extraction.**

care professionals, and persons with lived experience. The mapping analysis will occur through Microsoft excel. In the concept analysis, we will explore the current use and meaning of brain fog, by identifying its attributes, antecedents, and potential interventions [24]. The concept analysis will occur through the qualitative analysis software, Quirkos [1]. Both approaches to the data will rely on deductive qualitative content analysis techniques including in-depth topical summaries, as well as frequency reports to illustrate the prevalence of each theme identified during data analysis. To minimize bias and improve the reliability, credibility, and quality of findings data will be independently analyzed by two researchers.

## Implications

As described, the results of the proposed discourse analysis will assist in improving the overall understanding chronic pain related brain fog by integrating knowledge from social discussions with existing scientific literature. It is anticipated that this will generate social impacts to 1) help healthcare professionals better understand the lived experience of chronic pain related brain fog and 2) identify areas of exploration for future research.

Understanding how chronic pain related brain fog is discussed in social discourses will help healthcare professionals identify how to adapt treatments to best support persons with lived experiences [13,15]. To support healthcare professionals in identifying brain fog and adapting treatment, the proposed study aims to refine a definition of brain fog that incorporates information from both academic and lay discourses. Additionally, the results of the proposed study may identify any common sources of misinformation or miscommunication, which may act as a barrier towards clinician-patient rapport [16,17].

Further, the results of this study will help researchers account for brain fog within their studies, by providing a definition supported by both academic sources and persons with lived experiences. The results may also help researchers develop educational resources for both healthcare professionals and persons with lived experiences to promote a better social understanding of chronic pain related brain fog. Lastly, understanding how persons with lived experiences describe their experiences with brain fog and their unique concerns, will generate avenues of future research to improve the overall understanding of this phenomenon.

## Conclusion

To conclude, by incorporating an innovative and current methodology, this proposed discourse analysis of social media sources will generate nuanced insights, bridging the gap between the academic and lay community.

## Supporting information

**S1 File.**
(DOCX)

## Author Contributions

**Conceptualization:** Ronessa Dass, Tara Packham.

**Methodology:** Ronessa Dass, Tara Packham.

**Supervision:** Tara Packham.

**Writing – original draft:** Ronessa Dass, Tara Packham.

**Writing – review & editing:** Ronessa Dass, Tara Packham.

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
