## [Decision Letter · Decision Letter 0]

3 Dec 2023

PONE-D-23-23957Brain fog in chronic pain: Protocol for a discourse analysis of social media postingPLOS ONE

Dear Dr. Dass,

Thank you for submitting your manuscript to PLOS ONE. After careful consideration, we feel that it has merit but does not fully meet PLOS ONE’s publication criteria as it currently stands. Therefore, we invite you to submit a revised version of the manuscript that addresses the points raised during the review process.

One reviewer has expressed concerns about the methodology's reliance on social media and forums without scientific evidence, questioning its viability. The other reviewer recommended major revisions for the manuscript, particularly in the protocol, focusing on detailed aspects of data collection, annotation, and analysis, including transparency and rigor in qualitative research by involving multiple analysts for enhanced reliability and credibility.

We look forward to receiving your revised manuscript.

Kind regards,

Rashid Mehmood, PhD

Academic Editor

PLOS ONE

Journal Requirements:

3. Please amend your manuscript to include your abstract after the title page.

Reviewers' comments:

Reviewer's Responses to Questions

**Comments to the Author**

1. Does the manuscript provide a valid rationale for the proposed study, with clearly identified and justified research questions?

Reviewer #1: Yes

Reviewer #2: Partly

2. Is the protocol technically sound and planned in a manner that will lead to a meaningful outcome and allow testing the stated hypotheses?

Reviewer #1: Partly

Reviewer #2: Partly

3. Is the methodology feasible and described in sufficient detail to allow the work to be replicable?

Reviewer #1: No

Reviewer #2: No

4. Have the authors described where all data underlying the findings will be made available when the study is complete?

Reviewer #1: No

Reviewer #2: No

5. Is the manuscript presented in an intelligible fashion and written in standard English?

Reviewer #1: Yes

Reviewer #2: Yes

6. Review Comments to the Author

You may also provide optional suggestions and comments to authors that they might find helpful in planning their study.

Reviewer #1: Much of the information found on social media and forums may not be based on scientific evidence, making it difficult to differentiate between anecdotal reports and verified research findings. Therefore, I am not satisfied this is a viable methodology.

Reviewer #2: This manuscript requires a major review.

Please find below what needs to be included in the Protocol for it to be publishable.

Data collection:

You need to list

• Whether you are accessing the Twitter and Facebook public stream?

• How do you access Facebook data if profiles are closed?

• Will go to specific brain fog sites, will you become a member of these sites?

• What package you are using to access Twitter stream (.i.e. python)?

• What is the anticipate data for collection period?

• What are the explicit search terms are going to be used?

• Data collection duration? How many days/wks/months will you be collecting data?

• Will English language filters be applied?

• Will you exclude retweets to maintain originality

• Will keep information on the tweet/fb post data attribution (full-length tweet text, tweet ID, creation time, and Twitter user information)?

Annotation:

• Will you train classifiers to distinguish between personal and health-care related tweets.

• Will you manually label all tweets?

• Will you use the content of the tweet's full text for classification or just part of?

• Will you preprocess the text of the tweets by normalizing all URLs to one consistent string, removing special characters and English part of speech, converting all of the text to lowercase, and lemmatization to remove noise?

• Will you look at the top 1000 occurring terms (excluding common English stop words) and manually checked if the terms were relevant to health; wellness; diseases; side effects; conditions; body parts; and/or references to other substances against standard English, medical, and slang dictionaries?

Analysis:

• What tool will you use to undertake qualitative content analysis?

• For transparency and rigor in qualitative research, it's conventional and widely recommended to have two or more individuals independently analyse the data. This practice serves multiple valuable purposes:

• Minimizing Bias: When two or more researchers independently analyse the same qualitative data, it helps mitigate individual biases. Each analyst brings their unique perspective and experiences, and by having multiple analysts, you reduce the risk of one person's biases significantly influencing the interpretation of the data.

• Enhancing Reliability: Independent analysis by multiple individuals improves the reliability of the findings. It allows for the assessment of inter-coder reliability or inter-rater reliability, which measures the degree of agreement among coders or analysts. A high level of agreement suggests greater confidence in the validity of the findings.

• Quality Assurance: The collaborative approach helps in identifying and resolving discrepancies or disagreements in coding and interpretation. This iterative process can lead to more robust and well-supported findings.

• Richer Insights: Different analysts may notice unique patterns, themes, or nuances within the data. Multiple perspectives can lead to a deeper and more comprehensive understanding of the qualitative material.

• Enhancing Credibility: In qualitative research, demonstrating rigor and transparency is crucial for establishing credibility and trustworthiness. Engaging multiple analysts and documenting their consensus-building process can enhance the credibility of your research findings.

You need to include all the above in your protocol.

7. PLOS authors have the option to publish the peer review history of their article (what does this mean?). If published, this will include your full peer review and any attached files.

Reviewer #1: No

Reviewer #2: No

---

## [Author Response · Author response to Decision Letter 0]

13 Dec 2023

Thank you to the reviewers for their insightful feedback. We have done our best to address all comments point by point. 

Reviewer #1: Much of the information found on social media and forums may not be based on scientific evidence, making it difficult to differentiate between anecdotal reports and verified research findings. Therefore, I am not satisfied this is a viable methodology.

Thank you for highlighting your concerns. We acknowledge the limitations of social media posts in our introduction and will be sure to reiterate this point once we present our findings.

“Though social media data may be inherently biased, and at times inaccurate, the discourse often has real life implications (14,15). Persons with chronic pain may adopt the information they see online and may be influenced to alter their beliefs about their conditions (14,15).” 

However, our objective for this study is not to describe what has been stated in the academic literature, but rather we are interested in understanding the discourse of brain fog by persons with lived experiences. This is particularly important considering that the term brain fog was first derived through online discussions of persons with lived experiences. In our final manuscript, we will report how people describe this phenomenon. This information will be used to provide recommendations for future areas of research. 

Reviewer #2: This manuscript requires a major review.

Please find below what needs to be included in the Protocol for it to be publishable.

Data collection:

You need to list

• Whether you are accessing the Twitter and Facebook public stream?/How do you access Facebook data if profiles are closed/Will go to specific brain fog sites, will you become a member of these sites?/What package you are using to access Twitter stream (.i.e. python)?

In accordance with published ethical guidance, we have specified this information as follows:

In accordance the Internet Specific Ethical Questions Framework to protect data anonymity (22), searches will occur through a public search engine, therefore no private posts, pages, or groups will be searched.

We will acknowledge this approach as a limitation in the resulting manuscript, as we may indeed miss relevant material in closed Facebook groups.

• What is the anticipate data for collection period?/Data collection duration? How many days/wks/months will you be collecting data

We started data collection after submitting our protocol. Thus, data collection occurred in October 2023 within the span of one week. Data was collected within one week to ensure feasibility as new posts generate daily on social media platform. We have specified this information at the bottom of the search strategy paragraph.

• What are the explicit search terms are going to be used?

We have specified the search terms “#brainfog chronic pain,” “brain fog chronic pain,” “#brainfog chronic pain” and “brain fog #chronicpain”

• Will English language filters be applied?

No language filters will be applied, we have added this detail in the search strategy. 

• Will you exclude retweets to maintain originality

Thank you for highlighting this point, we have added it to our exclusion criteria.

• Will keep information on the tweet/fb post data attribution (full-length tweet text, tweet ID, creation time, and Twitter user information)?

All extracted potential identifiers will be deleted post manuscript completion, this information has been added at the end of the data extraction paragraph.

Annotation:

• Will you train classifiers to distinguish between personal and health-care related tweets/• Will you manually label all tweets?/ Will you use the content of the tweet's full text for classification or just part of?/Will you preprocess the text of the tweets by normalizing all URLs to one consistent string, removing special characters and English part of speech, converting all of the text to lowercase, and lemmatization to remove noise?/Will you look at the top 1000 occurring terms (excluding common English stop words) and manually checked if the terms were relevant to health; wellness; diseases; side effects; conditions; body parts; and/or references to other substances against standard English, medical, and slang dictionaries?

We will not be using artificial intelligence for this study, thus no classifiers or preprocessing. All searching and extraction will occur manually. We have specified this detail throughout the protocol. 

Analysis:

• What tool will you use to undertake qualitative content analysis?

The mapping analysis will occur on excel and the concept analysis will be done using the qualitative software, Quirkos. This detail has been added in the data analysis paragraph on page 4.

• For transparency and rigor in qualitative research, it's conventional and widely recommended to have two or more individuals independently analyse the data. This practice serves multiple valuable purposes: Minimizing Bias: When two or more researchers independently analyse the same qualitative data, it helps mitigate individual biases. Each analyst brings their unique perspective and experiences, and by having multiple analysts, you reduce the risk of one person's biases significantly influencing the interpretation of the data. Enhancing Reliability: Independent analysis by multiple individuals improves the reliability of the findings. It allows for the assessment of inter-coder reliability or inter-rater reliability, which measures the degree of agreement among coders or analysts. A high level of agreement suggests greater confidence in the validity of the findings. Quality Assurance: The collaborative approach helps in identifying and resolving discrepancies or disagreements in coding and interpretation. This iterative process can lead to more robust and well-supported findings. Richer Insights: Different analysts may notice unique patterns, themes, or nuances within the data. Multiple perspectives can lead to a deeper and more comprehensive understanding of the qualitative material. Enhancing Credibility: In qualitative research, demonstrating rigor and transparency is crucial for establishing credibility and trustworthiness. Engaging multiple analysts and documenting their consensus-building process can enhance the credibility of your research findings. You need to include all the above in your protocol.

Thank you for highlighting this! We have stated this at the end of the data analysis paragraph: 

To minimize bias and improve the reliability, credibility, and quality of findings data will be independently analyzed by two researchers.

---

## [Decision Letter · Decision Letter 1]

21 Feb 2024

PONE-D-23-23957R1Brain fog in chronic pain: Protocol for a discourse analysis of social media postingPLOS ONE

Dear Dr. Dass,

Thank you for submitting your manuscript to PLOS ONE. After careful consideration, we feel that it has merit but does not fully meet PLOS ONE’s publication criteria as it currently stands. Therefore, we invite you to submit a revised version of the manuscript that addresses the points raised during the review process. Please address the reviewers' concerns. Particularly, provide a more robust response to the comments by Reviewer 1 (add some text to the revised version supported by appropriate references). Consider PLOS ONE’s publication criteria in revising your manuscript.

We look forward to receiving your revised manuscript.

Kind regards,

Rashid Mehmood, PhD

Academic Editor

PLOS ONE

Reviewers' comments:

Reviewer's Responses to Questions

**Comments to the Author**

1. Does the manuscript provide a valid rationale for the proposed study, with clearly identified and justified research questions?

Reviewer #2: Yes

2. Is the protocol technically sound and planned in a manner that will lead to a meaningful outcome and allow testing the stated hypotheses?

Reviewer #2: Yes

3. Is the methodology feasible and described in sufficient detail to allow the work to be replicable?

Reviewer #2: Yes

4. Have the authors described where all data underlying the findings will be made available when the study is complete?

Reviewer #2: No

5. Is the manuscript presented in an intelligible fashion and written in standard English?

Reviewer #2: Yes

6. Review Comments to the Author

You may also provide optional suggestions and comments to authors that they might find helpful in planning their study.

Reviewer #2: Thank you for addressing the comments provided in the first review, well done I have provided some additional comments in the attached pdf. Good luck with you research.

7. PLOS authors have the option to publish the peer review history of their article (what does this mean?). If published, this will include your full peer review and any attached files.

Reviewer #2: No

---

## [Author Response · Author response to Decision Letter 1]

23 Feb 2024

Thank you to the reviewers for taking the time to review this article and provide thoughtful feedback. For simplicity, we have responded to the comment requiring additional information. 

4. Have the authors described where all data underlying the findings will be made available when the study is complete?

Reviewer #2: No

Thank you for highlighting this! Since we are using sources from social media, underlying findings will not be made available to maintain the anonymity of sources (22). We have specified this at the end of the data extraction section. 

Thank you for your time and please feel free to let us know if you have additional questions.

---

## [Editor Report · Decision Letter 2]

4 Apr 2024

Brain fog in chronic pain: Protocol for a discourse

analysis of social media posting

PONE-D-23-23957R2

Dear Dr. Dass,

We’re pleased to inform you that your manuscript has been judged scientifically suitable for publication and will be formally accepted for publication once it meets all outstanding technical requirements.

Kind regards,

Rashid Mehmood, PhD

Academic Editor

PLOS ONE
---

## [Editor Report · Acceptance letter]

26 Apr 2024

PONE-D-23-23957R2 

PLOS ONE

Dear Dr. Dass, 

I'm pleased to inform you that your manuscript has been deemed suitable for publication in PLOS ONE. Congratulations! Your manuscript is now being handed over to our production team.

Kind regards, 

on behalf of

Prof. Rashid Ibrahim Mehmood 

Academic Editor

PLOS ONE